



# $^{226}$Ra and $^{228}$Ra in the stratified estuary of the Krka River (Adriatic Sea, Croatia): implications for submarine groundwater discharge and its derived nutrients

Jianan Liu[1], Jinzhou Du[1], Blaženka Gašparović[2], Milan Čanković[2], Enis Hrustić[3], Neven

Cukrov[2], Zhuoyi Zhu[1], and Ruifeng Zhang[1, 4]

[1]State Key Laboratory of Estuarine and Coastal Research, East China Normal University, Shanghai 200062, PR China

[2]Division for Marine and Environmental Research, Ruder Bošković Institute, 10000 Zagreb, Croatia

[3]Institute for marine and coastal research, University of Dubrovnik, Kneza Damjana Jude 12, 20000
Dubrovnik, Croatia

[4]Institute of Oceanology, Shanghai Jiao Tong University, Shanghai 200240, PR China

*Correspondence to*: Jinzhou Du (jzdu@sklec.ecnu.edu.cn)

**Abstract**. We conducted a field survey in the Krka River and its estuary (Croatia) in September 2014 to study the significance of submarine groundwater discharge (SGD) and derived nutrients. During the sampling
period, the exchange time of the brackish water above the halocline in the Krka River Estuary (KRE) was calculated to be 8.3 ±3.5 days. Three approaches of three end-member model, mass balance model and time series observation in tidal period based on Ra isotopes were used to evaluate the SGD fluxes in the KRE surface layer. We estimated the SGD flux to be (1.3-7.8) $\times 10^5$ m$^3$ d$^{-1}$, which was approximately 2.7-16.1 % of the Krka River discharge into the estuary. By establishing the nutrient budgets in the KRE surface layer,
SGD dominated the nutrient sources, followed by Krka River. SGD-derived dissolved inorganic nitrogen (DIN) and dissolved orthosilicate (DSi) contributed 26.8-68.6 % and 9.5-38.3 % to the total DIN and DSi fluxes into the surface waters of KRE, respectively. This indicates that SGD was likely a major external source of those nutrients in the KRE. We have identified that SGD-derived nutrients and their high N:P ratios may affect the ecosystem productivity in the KRE and nearby Adriatic.

## 1. Introduction

A river-dominated estuary is the critical zone connecting the continent and adjacent sea. It is the primary pathway of freshwater and terrigenous materials transport to ocean through rivers. There are already numerous reports on the biogenic element processes in the estuaries and seas, for instance, trace metals, nutrients, carbon (e.g., Kelly and Moran, 2002; Hatje et al., 2003; Cai et al., 2004). More and more evidence
has indicated a significant transport of solutes via submarine groundwater discharge (SGD), which is defined as the flow of water from the seabed to coastal regions (e.g., Burnett et al., 2003). Radioactive isotopes have been well applied as tracers in evaluating SGD in coastal waters, especially radium isotopes ($^{223}$Ra, $^{224}$Ra,





[226]Ra and [228]Ra) that are widely used (e.g., Burnett et al., 2006; Moore et al., 2006). In addition, SGD-derived biogenic elements, especially nutrients, have the potential to impact the chemical budgets of the water ecosystems by changing the nitrogen to phosphorus (N:P) ratios, which may cause some environmental problems of eutrophication, harmful algal blooms and hypoxia (e.g., Lee et al., 2009; McCoy et al., 2011).

Further on, Kim et al. (2011) showed that SGD from two volcanic islands of Jeju was a very important external source of nutrients in the coastal oligotrophic ocean.

Being in area of the oligotrophic Adriatic Sea (the northernmost basin of the Mediterranean Sea) (Zavatarelli et al., 1998), the ecosystem of the Krka River Estuary (KRE) is very sensitive to the effluence of external substance input (Šupraha et al., 2014). Therefore, in this study we collected samples in Krka and its estuary

for Ra isotope and nutrient analyses, aiming to understand the importance of SGD in the KRE, not only in water balance, but also in SGD-derived nutrients that may affect the nutrient budgets and consequently system productivity.

## 2. Materials and methods

### 2.1. Study area

The Krka River is a typical groundwater-fed karstic river, situated on the eastern coast of the Adriatic Sea (Croatia) (Figure 1). The measured Krka River flow is between 5 and 565 $m^3$ $s^{-1}$ with an average annual flow between 40 and 60 $m^3$ $s^{-1}$ with rainy season in winter (Bonacci et al., 2006). Its hydrogeological drainage area covers an area of approximately 2427 $km^2$ and has a length of 49 km for the fresh water section, while the estuary is extended for additional 24 km. The KRE is a typical stratified estuary that was formed during

the Holocene transgression, with a fresh-brackish surface layer moving seawards and a bottom seawater layer as a countercurrent, moving upwards (Cukrov et al., 2009). The exchange time of freshwater in the KRE is between 6 and 20 days during winter, whereas for the marine water is from 50 to 100 days (Legović, 1991). Characteristic of the KRE is weak tidal amplitude of no more than 0.5 m (Žic and Branica, 2006), so, the Krka River plays a dominant role in controlling the water quality and permanent vertical stratification in the

estuary (e.g., Legović et al., 1994; Cukrov et al., 2012). The occurrence of numerous submerged or ephemeral springs along the Krka River (e.g., Kniewald et al., 2006; Cukrov et al., 2012), as one type of submarine groundwater, have the potential to affect the ecosystem by its associated substance transporting into the estuary. Besides that, with the rising intensity of anthropogenic activities, subtle changes occurred in the ecosystem of the Krka River and its estuary (e.g., Kwokal et al., 2002; Cukrov et al., 2008). In the lower

KRE under the effluence of municipal wastewaters from the city of Šibenik, increased concentrations of trace metals (e.g., Cindrić et al., 2015) and nutrients (e.g., Svensen et al., 2007) were detected.

### 2.2. Sampling strategy

A field survey was performed in the Krka River and its estuary during the period of September 4-10[th], 2014. At that time the average Krka River flow was 56 $m^3$ $s^{-1}$ (Figure 2). The sampling covered transect from the



upper stream of the Krka River via the estuary to the coastal water in the Adriatic Sea. Sampling locations for radium isotopes are given in Figure 1. Samples of surface waters were collected directly from a depth of approximately 0.5 m using the Niskin bottles to fill the container of 60 L per sample, whilst subsurface waters were taken also by Niskin bottles at a depth of approximately 2 m to fill the container of 20 L per sample. In the KRE, we also conducted a 24 hours-time series observation by sampling the surface water at time-series (TS) station every three hours (Figure 1). Groundwater samples were collected in springs and wells along the KRE. In addition, corresponding water samples for nutrient analysis were collected in polyethylene bottles. Vertical profiles of physico-chemical parameters of interest (salinity, temperature and dissolved oxygen (DO)) were measured in-situ at each site by multi parametric probe (Hach Lange HQ40D).

## 2.3. Methods

The water samples for Ra were passed through a column filled with about 20 g $MnO_2$-impregnated acrylic fiber at a flow rate of around 0.5 L min$^{-1}$ after the suspended sediments were removed by filtration cartridges (pore size of 0.5 mm). Radium isotopes $^{228}Ra$ and $^{226}Ra$ were determined by gamma spectrometry (Ortec, GWL-120-15-XLB-AWT) (Wang et al., 2014). Briefly, after leached from the $MnO_2$-impregnated acrylic fiber, Ra isotopes were co-precipitated with barium sulfate, and the precipitation was sealed for more than 20 days before measurement. The counting time for each sample was 24 to 48 h. The $^{226}Ra$ activities were measured using the $^{214}Pb$ (295 keV and 352 keV) and $^{214}Bi$ (609 keV) peaks; the $^{228}Ra$ activities were used the 338 keV and 911 keV peaks of $^{228}Ac$. The uncertainties of $^{226}Ra$ and $^{228}Ra$ were 1.56 - 20.20 % and 2.77 - 20.80 %, respectively.

Samples (50 mL) for the analysis of ammonium ($NH_4^+$) were stabilized by the addition (2 mL) of the phenol (1 mol L$^{-1}$; dissolved in 95 % vol/vol ethanol) (Ivančić and Degobbis, 1984) and stored in the dark at 4 ℃. The samples (500 mL) for other nutrients were stored at -22 ℃. The concentrations of nitrate ($NO_3^-$), nitrite ($NO_2^-$), $NH_4^+$, reactive orthosilicates ($SiO_4^{4-}$), hereafter termed DSi, orthophosphate ($PO_4^{3-}$, hereafter termed DIP), were determined after Strickland and Parsons (1972). The concentration of DIN is the sum of the $NO_2^-$, $NO_3^-$, and $NH_4^+$ concentrations.

## 3. Results and discussion

### 3.1. Hydrological features

During the investigated period, salinity in the surface water ranged from 0.2 to 33.3 for stations KR2 to KR10. Freshwater affected only upper ~2.5 m, thickness of which decreased approaching to the seaside (Figure 3a). The maxima of temperature and DO concentration were detected along the edge of halocline throughout the investigated estuary. That is explained by the solar radiation passage through the transparent brackish water and a slow entrainment in the marine water (Legović et al., 1994). Temperature and DO decreased gradually from the halocline towards the bottom water (Figures 3b and 3c). The halocline divided the water column into two parts. The freshwater-seawater interface (halocline) was described as a filter that makes two layers



different not only hydrologically, but also chemically and biologically (e.g., Legović et al., 1994; Svensen et al., 2007; Žic and Branica, 2006).

### 3.2. Dissolved nutrients in the Krka River and its estuary

Nutrient concentrations ($\mu$mol $L^{-1}$) ranged from 1.00 to 5.81 for DIN, 0.21 to 0.69 for DIP and 3.36 to 32.92

for DSi in the surface water of the Krka River and the KRE (Table 1). The DIN and DSi concentrations decreased from the upper stream towards the mouth of the estuary and had a significant negative correlation with salinity (Figures 4b and 4d). No obvious relationship between DIP concentration and salinity was observed (Figure 4c). The groundwater around the KRE had higher DIN and DSi than the Krka River water, but similar concentration of DIP. DIN/DIP i.e. N:P ratios in the water columns at most of the stations were

below the Redfield ratio of 16 (Redfield et al., 1963) indicating a potential lack of nitrogen for the balanced growth of phytoplankton. In addition, high Ra activities corresponded to high nutrient concentrations (Figure 4), indicating that SGD input was associated with high Ra and nutrient values. DIN and DSi concentrations increased with the depth up to the halocline in the upper water, and decreased in the underlying water below the halocline, while DIP showed opposite trend with minimum near the halocline (Figure 5).

### 3.3. Dissolved Ra in the Krka River

The dissolved $^{226}$Ra and $^{228}$Ra activities during the sampling period ranged from 86 to 130 dpm $m^{-3}$ and from 33 to 127 dpm $m^{-3}$, respectively (Figure 4a and Table 1). The measured values were lower than those observed in some other Croatian rivers (Bituh et al., 2008). The $^{226}$Ra activity in Krka River was similar to the activity measured in some rivers worldwide, while $^{228}$Ra activity was lower in the Krka River (e.g. Krest et al., 1999;

Su et al., 2015). The reason may be that the carbonate sedimentation prevails in this karst river, resulting in U (a parent of $^{226}$Ra) becomes more enriched than Th (a parent of $^{228}$Ra) (Cukrov et al., 2009; Cukrov and Barišić, 2006). Ra activities were low in the fresher waters in the upper KRE, the highest at the KRE mouth and again low out of the estuary (Figure 4a). There are several underground springs that flow out into the KRE, especially in the bay near the Zaton village in the lower part of the estuary (N. Cukrov, personal comm.),

that may contribute to higher Ra activities of the stations KR4-KR8.

### 3.4. Time series observation

The results from 24 hours-time series observation conducted in the estuary showed that all the parameters varied over a tidal cycle even though the tidal range was low, less than 0.3 m. Salinity varied consistently with the tidal height. Maximum and minimum salinities were found at high and low tide, respectively, and

were in a range from 10.7 to 14.0 (Figure 6 and Table 1). The $^{226}$Ra and $^{228}$Ra activities over the time series observation ranged from 91 to 119 dpm $m^{-3}$ with an average of 106 $\pm$15 dpm $m^{-3}$ and from 63 to 139 dpm $m^{-3}$ with an average of 92 $\pm$22 dpm $m^{-3}$, respectively. Activities of $^{226}$Ra and $^{228}$Ra showed an opposite trend with respect to salinity because of the dilution with the open seawater that has low Ra activity (Figures 6a



and 6b). Similar patterns were also observed in other places around the world (Garcia-Orellana et al., 2010; Wang et al., 2016). During the lower salinity period, the $^{226}$Ra and $^{228}$Ra activities were higher although some deviation due to hysteresis effect was observed. This was the result of more fresh water and submarine groundwater that come out bringing higher Ra activities.

Nutrient concentrations also varied with the salinity changes during the time series observation (Figure 6c). High DIN and DSi concentrations occurred in lower salinity waters. Similar to that of Ra activity, DIN and DSi variations had an opposite trend to salinity despite a small hysteresis effect observed (Figure 6c). There was no obvious variation trend between DIP and salinity. Overall, both Ra activity and nutrient concentrations varied over the time series observation. The variations were on account of different material sources, including those from open seawater, river water and from SGD.

### 3.5. Three end-member mixing model

Plots of $^{226}$Ra and $^{228}$Ra activities versus salinity showed that the $^{226}$Ra and $^{228}$Ra activities in the surface estuarine water were higher than those expected from a conservative mixing line between Krka River water and open seawater (Figure 7), indicating that there was an excess of Ra entering the estuary through other sources, such as SGD (e.g., Peterson et al., 2008; Moore, 2010). It was particularly pronounced for $^{228}$Ra that had lower effect than in the open Adriatic Sea. Therefore, a three end-member mixing model was based on salinity and $^{228}$Ra to estimate the fractions of (1) open seawater, (2) river water and (3) groundwater in the KRE surface waters.

We used equations for water, salinity and $^{228}$Ra balance as follows (Moore, 2003):

$$f_S + f_R + f_{GW} = 1.00 \tag{1}$$

$$S_S f_S + S_R f_R + S_{GW} f_{GW} = S_M \tag{2}$$

$$^{228}Ra_S f_S + {}^{228}Ra_R f_R + {}^{224}Ra_{GW} f_{GW} = {}^{228}Ra_M \tag{3}$$

where f refers to the fraction of the open seawater (S), river (R) and groundwater (GW) end-member; $S_S$, $S_R$, $S_{GW}$ and $^{228}Ra_S$, $^{228}Ra_R$, $^{228}Ra_{GW}$ are the salinity and $^{228}$Ra activity in the open seawater, river and groundwater, respectively. The subscript M represents the measured value for the salinity and $^{228}$Ra of individual sample.

Then equations above can be solved for the fraction of each end-member:

$$f_S = \frac{\left(\dfrac{^{228}Ra_M - {}^{228}Ra_R}{^{228}Ra_{GW} - {}^{228}Ra_R}\right) - \left(\dfrac{S_M - S_R}{S_{GW} - S_R}\right)}{\left(\dfrac{^{228}Ra_S - {}^{228}Ra_R}{^{228}Ra_{GW} - {}^{228}Ra_R}\right) - \left(\dfrac{S_S - S_R}{S_{GW} - S_R}\right)} \tag{4}$$

$$f_{GW} = \frac{S_M - S_R - f_S(S_S - S_R)}{S_{GW} - S_R} \tag{5}$$

$$f_R = 1.00 - f_S - f_{GW} \tag{6}$$



With the three end-member values shown in Figure 7, we evaluated the fractions of open seawater, river water and groundwater in the KRE surface layer waters. The model results are shown in Figure 8. As expected in the surface layer of the KRE, the fraction of the river water was higher than those of the open seawater and groundwater. During the time series observation, lower changes (28-37 %) were observed for the open seawater fraction relative to the values of the river water and groundwater.

### 3.6. Flushing time in the surface layer of KRE

As KRE is highly stratified we were interested in computing flushing time mainly for the surface fresh and brackish layer which was approximately 2.5 m deep. We used a method based on physical model described by Moore et al. (2006) as follows:

$$T_f = \frac{VT}{(1-b)P} \tag{7}$$

here $T_f$ is the flushing time, V refers to the volume of the surface estuarine water which is defined as the product of the average area and depth, T is the tidal period, P is the tidal prism and b represents the return flow into the open sea from the study region. In the investigated estuary, regular semidiurnal tidal period equals approximately 0.47 days from the time series observation. The tidal prism can be determined by multiplying the average surface area by the tidal range during the sampling period, which we estimated to be $2.6 \times 10^6$ m$^3$. In this model, b is equivalent of the open seawater fraction, whilst the fraction of open seawater represents only the surface water. So, based on the salinity profiles of the KRE surface layer waters, we calculated the fraction of open seawater in the total surface layer water of the KRE to be $0.49 \pm 0.21$. Therefore, the flushing time of KRE surface layer water was estimated to be $8.3 \pm 3.5$ days, which is comparable to the value of 10 days in September reported by Legović (1991).

### 3.7. Estimation of SGD

### 3.7.1. SGD derived from a three end-member mixing model

Based on the three end-member mixing model, we also calculated the faction of groundwater in the estuary to be $0.21 \pm 0.07$. Assuming that this value represents the fraction of groundwater in the total KRE surface layer, we can obtain the flux of SGD using the following equation:

$$SGD = \frac{V f_{GW}}{T_f} \tag{8}$$

Therefore, the flux of SGD into the KRE surface layer was calculated to be $(2.6\text{-}9.1) \times 10^5$ m$^3$ d$^{-1}$.

### 3.7.2. Ra mass balance model for estimating SGD

Generally, in a defined system with an assumed steady state, the Ra mass balance is equal to the sum of inputs, which are usually from river supply, sediments diffusion, SGD, and the loss, which includes open





seawater mixing and Ra decay (e.g., Moore et al., 2006). Based on this, Ra mass balance model is another approach to quantify the magnitude of SGD. This model has been widely used in estuaries around the world (e.g., Moore et al., 2006; Rengarajan and Sarma, 2015). We carried out a mass balance model of $^{228}$Ra to estimate the SGD flux in the KRE surface layer (Figure 9). The existence of the halocline in the KRE will

prevent the $^{228}$Ra diffusion from the sediments through the halocline into the surface layer water, so we ignored the term of sediments diffusion. The atmospheric

We can write the following Eq. (9) for the $^{228}$Ra mass balance model for the KRE surface layer:

$$F(^{228}Ra_{river}) + F(^{228}Ra_{SGD}) = F(^{228}Ra_{mix}) \tag{9}$$

where the subscript river and SGD represent the Ra fluxes input by Krka River and SGD, respectively;

the subscript mix represents the Ra loss by mixing with open seawater. Then SGD-derived $^{228}$Ra flux and SGD flux in the KRE surface layer can be given by Eq. (10):

$$F(^{228}Ra_{SGD}) = [C(^{228}Ra_{ES}) - f_S \times C(^{228}Ra_{SW})] \times A_{ES} \times H_{upper} \times (1/T_f) - C(^{228}Ra_{river}) \times F_{river} \tag{10}$$

$$Q_{SGD} = \frac{F(^{228}Ra_{SGD})}{^{228}Ra_{gw}} \tag{11}$$

where $C(^{228}Ra_{ES})$, $C(^{228}Ra_{SW})$ and $C(^{228}Ra_{river})$ are the activities of $^{228}$Ra in the KRE surface layer, open

seawater and the Krka River, respectively. $A_{ES}$ and $H_{upper}$ are the area and depth of the KRE surface layer (2.5 m). $T_f$ is the measured flushing time and $F_{river}$ is the Krka River flow during the sampling period. Based on Eqs. (10) and (11), we determined the SGD flux in the KRE surface layer to be $(1.6–6.0) \times 10^5$ m$^3$ d$^{-1}$ with all the parameters summarized in Table 2.

### 3.7.3. SGD derived from the tidal cycles

We also used another method based on the Ra time series observation of tidal cycles to estimate the SGD flux in the KRE surface layer. Following the approach of Peterson et al. (2008), Eq. (12) was used to estimate the SGD flux as follows:

$$Q_{SGD} = \frac{(Ra_{total} - Ra_{bkgd}) \times A_{ES} \times H_{upper}}{T_f \times Ra_{gw}} \tag{12}$$

Here we used the following steps to evaluate the SGD flux:

i) As the each measured Ra activity ($Ra_{total}$) in the KRE surface layer was the result of total Ra source, we calibrated the each measured Ra activity by subtracting out the estuarine background Ra activity ($Ra_{bkgd}$). We chose the minimum activity from the measured values of the time series observation as the background of the estuarine water with a conservative SGD estimation. In this way, we could conclude that the excess Ra activity exclusively came from SGD.





ii) Assuming that the Ra activity of time series observation can represent the Ra activity in the total KRE surface layer, we estimated the excess Ra inventory by multiplying by excess Ra activity with the KRE surface layer depth ($H_{upper}$, 2.5m) and the estuary area ($A_{ES}$, $9.3 \times 10^6$ m$^2$).

iii) The excess Ra inventory can be converted to Ra flux by dividing with the estimated flushing time of the surface estuarine water ($T_f$, 8.3 days).

iv) Finally, by dividing the Ra flux with the Ra activity in the groundwater end-member ($Ra_{gw}$), which was $316 \pm 24$ dpm m$^{-3}$ for $^{228}$Ra, then we could obtain the SGD flux in the KRE surface layer.

Similar to the above calculation, here we only chose $^{228}$Ra to estimate the SGD flux because of its lower activity in the open seawater. Based on the Ra activities of time series observation and by applying Eq. (12), we were able to determine the SGD flux for each time series sample. The range of SGD fluxes in the KRE surface layer during the tidal cycles were estimated to be $(1.2–6.8) \times 10^5$ m$^3$ d$^{-1}$.

Therefore, the SGD flux in the upper Krka estuary was estimated to be $(2.6–9.1) \times 10^5$, $(1.6–6.0) \times 10^5$ and $(1.2–6.8) \times 10^5$ m$^3$ d$^{-1}$ by three end-member mixing model, $^{228}$Ra mass balance model and time series observation, respectively. The calculated results of these three approaches were similar and in a reasonable agreement, which gives us a confidence that the estimation of SGD flux in the KRE surface layer was grounded, being in the range of $(1.3–7.8) \times 10^5$ m$^3$ d$^{-1}$. In this way, the amount of SGD is accounted to be 2.7-16.1 % of the Krka River discharge into the KRE surface waters during the sampling period. By comparing with the other studies in the Mediterranean region (Table 3), we can see that the estimated SGD flux from this study is comparable to other reported values.

In addition, we followed Wang et al. (2015) to evaluate a water mass balance in the KRE surface layer under the assumption that the study area of interest was a single box at steady state. The conceptual water mass balance for the KRE surface layer is presented by Figure 10. The total water inflow should be precipitation ($Q_P$) and freshwater discharge including the total river flux ($Q_R$), wastewater ($Q_W$) and $Q_{SGD}$. The water outputs included residual flow ($Q_O$) out of the KRE surface layer to the Adriatic Sea and the evaporation ($Q_E$). Thus the total water inflow equals to the water outflow, and the water mass balance can be written by the following equation:

$$Q_R + Q_P + Q_W + Q_{SGD} = Q_E + Q_O \qquad (13)$$

In this study, the precipitation and evaporation fluxes were $45.3 \times 10^5$ and $30.5 \times 10^5$ m$^3$ d$^{-1}$ (data from http://www.esrl.noaa.gov/). The wastewater that spreads out from the city of Šibenik has an average outflow of approximately $0.046 \times 10^5$ m$^3$ d$^{-1}$ (data from http://www.wte.de/WTE-Group.aspx). Then based on Eq. (13), the residual flow out of the KRE surface layer was estimated to be $67.8 \times 10^5$ m$^3$ d$^{-1}$. From the water mass balance (Figure 10), we can see that SGD contribution to the total water inflow was very small being only 1.3-8.0 %. In contrast, contribution from the Krka River was up to at least 48 %, which was the most important component. Besides, the water exchange flow or mixing flow between the KRE surface layer and the open sea ($Q_M$) can be derived based on the salt balance by following equation:

$$Q_M(S_2 - S_1) = Q_O(S_1 + S_2)/2 \qquad (14)$$



where $S_1$ (18.2) and $S_2$ (36.9) represent the mean salinities of the KRE surface layer and the Adriatic Sea, respectively. We estimated water exchange flow or water mixing flow to be $99.7 \times 10^5$ m$^3$ d$^{-1}$, which included water mixing flow out of the estuary and exchange with the underlying water below the halocline.

**3.8. Evaluation of SGD-derived nutrient fluxes to the KRE surface layer**

Nutrients in the KRE were thought to mainly originate from the freshwater inflow of Krka River and from anthropogenic sources near the city of Šibenik (e.g., Legović et al., 1994; Svensen et al., 2007). Here we determined that Krka River contributed to the estuary with $22.4 \times 10^3$ mol d$^{-1}$ for DIN, $1.3 \times 10^3$ mol d$^{-1}$ for DIP and $150 \times 10^3$ mol d$^{-1}$ for DSi during the sampling time. However, recently the SGD-derived nutrients have been shown as a major component and indisputable sources in some estuarine systems (e.g., Su et al., 2011; Rengarajan and Sarma, 2015). In the groundwater of the KRE, the nutrient concentrations were on average 103, 0.44 and 120 μmol L$^{-1}$ for DIN, DIP and DSi, respectively during the sampling period, in which the DIN and DSi concentrations were much higher than those in the Krka River and the estuarine water. Therefore, considering the groundwater sampled in a shallow depth, the SGD-derived nutrient fluxes to the KRE surface layer were estimated to be $(13.6–80.6) \times 10^3$, $(0.058–0.35) \times 10^3$ and $(15.8–93.8) \times 10^3$ mol d$^{-1}$ for DIN, DIP and DSi, respectively. The fluxes were equivalent to 60–360 %, 4.5–27 % and 10–63 % of the riverine inputs to the KRE surface layer for DIN, DIP and DSi, respectively. It seems that the SGD in the study region provides a substantial contribution to the DIN, DIP and DSi loadings to the Krka estuarine system.

Similar to the water balance, assuming that the study was conducted at a steady state, we established nutrient budgets in the KRE surface layer. It is based on a box model devised by Land Ocean Interactions in the Coastal Zone (Gordon et al., 1996), which has been widely used to evaluate the relative importance of external nutrient inputs versus the physical transports and internal biogeochemical processes within a body of water (e.g., Liu et al., 2009, 2011; Wang et al., 2016). Except from river and SGD inputs, atmospheric deposition and wastewater were the other two sources for nutrient input to the KRE surface layer. These sources can be estimated by multiplying atmospheric deposition rate with the surface area (Markaki et al., 2010; Rodellas, 2015) and multiplying wastewater nutrient concentrations with the wastewater flux (Gunes et al., 2012; Powley, et al., 2016), respectively. In terms of nutrient outputs in this model, the net residual flux had a significant role. It can be estimated by $Q_O \times (C_1 + C_2)/2$, where $C_1$ and $C_2$ are the nutrient concentrations in the KRE surface layer and the open seawater, respectively. Another term of nutrient fluxes out of the upper Krka estuary is the exchange with the open seawater, here obtained as $Q_M \times (C_1 - C_2)$. Therefore, based on the above estimated results of each nutrient flux, the nutrient budgets in the KRE surface layer were shown in Figure 11. We found that the amount of nutrient inputs were greater than the outputs, showing that the KRE surface layer system was a sink of nutrients. Nutrients could be deposited down to the underlying water, taken up by the plankton community and lost by outflow to the Adriatic Sea. SGD was the dominant source of DIN and DSi, which contributed 26.8–68.6% and 9.5–38.3 % to the total DIN and DSi fluxes into the KRE surface layer, respectively, followed by Krka River and wastewater.



Generally, N:P ratios in groundwater are greater than the requirements of phytoplankton growth (16:1) (Slomp and Van Cappellen, 2004), and it was widely observed that SGD-derived N:P ratios were much higher than those from the rivers and other sources in coastal waters (e.g., Hwang et al., 2005; Lee et al., 2009; Waska and Kim, 2011). In this study, an average N:P ratio of SGD-derived nutrients was approximately 233, much higher than those found in the Krka River (~17) and wastewater from the city of Šibenik (~12). Considering the large amount of DIN and DIP in the SGD, it might be significant in changing the nutrient structure of the KRE surface layer. This corresponded to the environmental problems such as eutrophication (Legović et al., 1994), phytoplankton bloom (Petricioli et al., 1996; Svensen et al., 2007) and hypoxia (e.g., Legović et al., 1991; Cetinić et al., 2006) in the KRE, especially in the Zaton Bay region, where the water exchange time is relatively long, resulting in frequent occurrence of the red tides. Surely, the SGD with high N:P ratios may have a significant effect on the ecosystem of the KRE region, which should be further investigated in the future. Moreover, supply of DIN, DIP and DSi to the studied area through SGD was a significant contributor to the overall nutrient budgets.

## 4. Conclusions

In the highly stratified estuary of the Krka River, we used three approaches including three end-member mixing model, mass balance model and time series observation in tidal periods to evaluate the SGD in the upper water above the halocline. Based on $^{226}$Ra and $^{228}$Ra activities, the SGD flux was estimated to be $(1.3–7.8) \times 10^5$ m$^3$ d$^{-1}$ in the KRE surface layer. Even if SGD accounted for a small portion of the total water in the study area relative to the Krka River discharge and precipitation, nutrient fluxes through SGD were significant sources for the nutrient budget in the KRE surface layer, especially for DSi. Nevertheless, nutrient-enriched SGD with high N:P ratios have the notable potential to impact the ecosystem of the KRE. In the further work, this should be investigated in more details.

### Author contribution

J. Du and B. Gašparović designed the experiment and sampling strategy and J. Du, B. Gašparović, Z. Zhu and R. Zhang carried them out; J. Liu, M. Čanković, E. Hrustić and N. Cukrov analyzed the samples; J. Liu prepared the manuscript with contributions from all co-authors.

### Competing interests

The authors declare that they have no conflict of interest.

### Acknowledgements

Financial support of the Croatian-Chinese Scientific and Technological Cooperation and Open Research Fund of State Key Laboratory of Estuarine and Costal Research (Grant number SKLEC-KF201505) and the





Natural Science Foundation of China (Grant number 41376089) are gratefully acknowledged. We thank all the colleagues in the field survey.

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



**Table 1** Activities of $^{226}$Ra and $^{228}$Ra and concentrations of nutrients in the Krka River and its estuary and groundwater samples during the sampling period.

| Sample | Sampling date | Layer | Salinity | $^{226}$Ra | Error | $^{228}$Ra | Error | DIN | PO$_4^{3-}$ | SiO$_4^{4-}$ |
|---|---|---|---|---|---|---|---|---|---|---|
| | mm/dd/yy | m | | dpm m$^{-3}$ | | | | μmol L$^{-1}$ | | |
| KR1 | 9/4/2014 | 0 | 0.2 | 90 | 4.9 | 33 | 7.0 | 4.63 | 0.27 | 31.0 |
| KR2 | 9/4/2014 | 0 | 2.3 | 89 | 5.3 | 61 | 7.7 | 4.16 | 0.54 | 26.9 |
| KR3 | 9/9/2014 | 0 | 7.1 | 86 | 4.9 | 57 | 7.2 | 5.40 | 0.69 | 32.9 |
| | | 1 | 7.6 | na | | na | | 3.47 | 0.13 | 38.3 |
| | | 2 | 14.8 | 127 | 11 | 114 | 10 | 5.77 | 0.16 | 29.4 |
| | | 6 | 35.3 | na | | na | | 0.63 | 0.63 | 8.30 |
| | | 12 | 35.5 | 122 | 8.1 | 131 | 9.4 | 0.57 | 0.19 | 4.57 |
| KR4 | 9/9/2014 | 0 | 11.2 | 101 | 5.0 | 102 | 6.7 | 5.81 | 0.24 | 25.8 |
| KR5 | 9/7/2014 | 0 | 20 | 116 | 4.6 | 102 | 6.5 | na | na | na |
| KR6 | 9/8/2014 | 0 | 14.9 | 91 | 5.1 | 126 | 6.4 | 4.75 | 0.33 | 22.4 |
| KR7 | 9/7/2014 | 0 | 19 | 116 | 5.0 | 115 | 7.0 | na | na | na |
| KR8 | 9/5/2014 | 0 | 21.7 | 130 | 4.6 | 127 | 7.0 | 2.96 | 0.24 | 13.4 |
| | | 2 | 32.5 | 134 | 9.6 | 163 | 9.4 | 1.43 | 0.18 | 5.28 |
| KR9 | 9/8/2014 | 0 | 27.1 | 109 | 5.0 | 82 | 7.6 | 2.10 | 0.21 | 8.61 |
| | | 2 | 33.4 | 103 | 9.3 | 385 | 11 | 0.78 | 0.26 | 3.26 |
| KR10 | 9/8/2014 | 0 | 33.3 | 103 | 5.6 | 86 | 8.2 | 1.89 | 0.25 | 6.29 |
| KR11 | 9/5/2014 | 0 | 36.9 | 107 | 5.3 | 49 | 7.7 | 1.00 | 0.25 | 3.36 |
| *Groundwater* | | | | | | | | | | |
| GW-1 | 9/6/2014 | na | 0.2 | 105 | 8.6 | 205 | 14 | na | na | na |
| GW-2 | 9/7/2014 | 0.5 | 3.0 | 966 | 9.3 | 427 | 20 | 8.12 | 0.34 | 120 |
| GW-3 | 8/30/2016 | 0.2 | 10.1 | na | na | na | na | 150 | 0.52 | na |
| GW-4 | 8/30/2016 | 0.2 | 22.4 | na | na | na | na | 152 | 0.47 | na |
| *Time series observation* | | | | | | | | | | |
| TS-1 | 9/9/14 10:40 | 0 | 11.6 | 102 | 5.0 | 63 | 7.9 | 5.77 | 0.23 | 31.1 |
| TS-2 | 9/9/14 13:40 | 0 | 14.0 | 108 | 5.1 | 85 | 7.8 | 3.88 | 0.24 | 23.6 |
| TS-3 | 9/9/14 16:30 | 0 | 13.9 | 103 | 5.1 | 91 | 7.5 | 4.09 | 0.39 | 23.2 |
| TS-4 | 9/9/14 19:30 | 0 | 13.2 | 91 | 4.8 | 96 | 6.2 | 3.56 | 0.39 | 27.9 |
| TS-5 | 9/9/14 22:00 | 0 | 12.3 | 117 | 5.2 | 119 | 7.0 | 3.97 | 0.31 | 29.9 |
| TS-6 | 9/10/14 1:40 | 0 | 11.0 | 106 | 4.9 | 79 | 7.8 | 4.09 | 0.36 | 28.6 |
| TS-7 | 9/10/14 4:40 | 0 | 11.0 | 114 | 4.9 | 81 | 7.8 | 6.57 | 0.31 | 26.9 |
| TS-8 | 9/10/14 7:40 | 0 | 11.2 | 119 | 5.0 | 139 | 7.2 | 3.09 | 0.33 | 30.5 |
| TS-9 | 9/10/14 10:40 | 0 | 10.7 | 93 | 5.1 | 76 | 7.1 | 4.69 | 0.59 | 32.0 |

na: not available





**Table 2** Definition and values used in the simultaneous equations for $^{228}$Ra mass balances for calculating SGD flux in the KRE surface layer.

| Parameter | Definition | Value | Unit |
|---|---|---|---|
| $^{228}$Ra$_{ES}$ | $^{228}$Ra activity in the KRE surface layer | 124 ±32 | dpm m$^{-3}$ |
| $^{228}$Ra$_{SW}$ | $^{228}$Ra end-member in the open seawater | 49 ±7.7 | dpm m$^{-3}$ |
| $^{228}$Ra$_{river}$ | $^{228}$Ra end-member in the Krka River water | 33 ±7.0 | dpm m$^{-3}$ |
| $^{228}$Ra$_{gw}$ | $^{228}$Ra end-member in the groundwater | 316 ±24 | dpm m$^{-3}$ |
| A$_{ES}$ | Surface area of the KRE | $9.3 \times 10^6$ | m$^2$ |
| H$_{upper}$ | Water depth of the KRE | 2.5 | m |
| T$_f$ | Measured flushing time in the KRE surface layer | 8.3 ±3.5 | d |
| F$_{river}$ | Krka River freshwater discharge | $4.8 \times 10^6$ | m$^3$ d$^{-1}$ |





**Table 3** A comparison of SGD fluxes around the Mediterranean region.

| Region | Tracer | SGD ($m^3\ d^{-1}$) | Percentage to river | Reference |
|---|---|---|---|---|
| Balearic Islands, Spain | [223,224,226,228]Ra | $411 \pm 118$ | na | Garcia-Solsona et al., 2010 |
| Pen ścola marsh, Spain | [222]Rn & [226]Ra | $(0.56\text{-}2.94) \times 10^5$ | na | Rodellas et al., 2012 |
| Messiniakos Gulf, Greece | [222]Rn | $(0.17\text{-}1.1) \times 10^5$ | 4-20 % | Pavlidou et al., 2014 |
| Donnalucata, Italy | [226]Ra | $5 \times 10^6$ | na | Moore, 2006 |
| | [222]Rn | $(1.2\text{-}7.4) \times 10^3$ | na | Burnett and Dulaiova, 2006 |
| Lesina Lagoon, Italy | [224]Ra | $(0.75\text{-}1.0) \times 10^6$ | 350-500 % | Rapaglia et al., 2012 |
| Venice Lagoon, Italy | [226]Ra | $(1.7\text{-}5.6) \times 10^7$ | na | Rapaglia et al., 2010 |
| Palma Beach, Spain | [226,228]Ra | $(5.6 \pm 1.3) \times 10^4$ | na | Rodellas et al., 2014 |
| Gulf of Lion, France | [226,228]Ra | $(0.24\text{-}4.5) \times 10^7$ | 1.6-29 % | Ollivier et al., 2008 |
| Marina Lagoon, Egypt | [222]Rn | $(0.83\text{-}2.4) \times 10^5$ | na | El-Gamal et al., 2012 |
| Mediterranean Sea | [228]Ra | $(0.82\text{-}13) \times 10^9$ | 100-1600 % | Rodellas et al., 2015 |
| Upper Krka estuary, Coratia | [228]Ra | $(1.3\text{-}7.8) \times 10^5$ | 2.7-16 % | This study |

na: not available





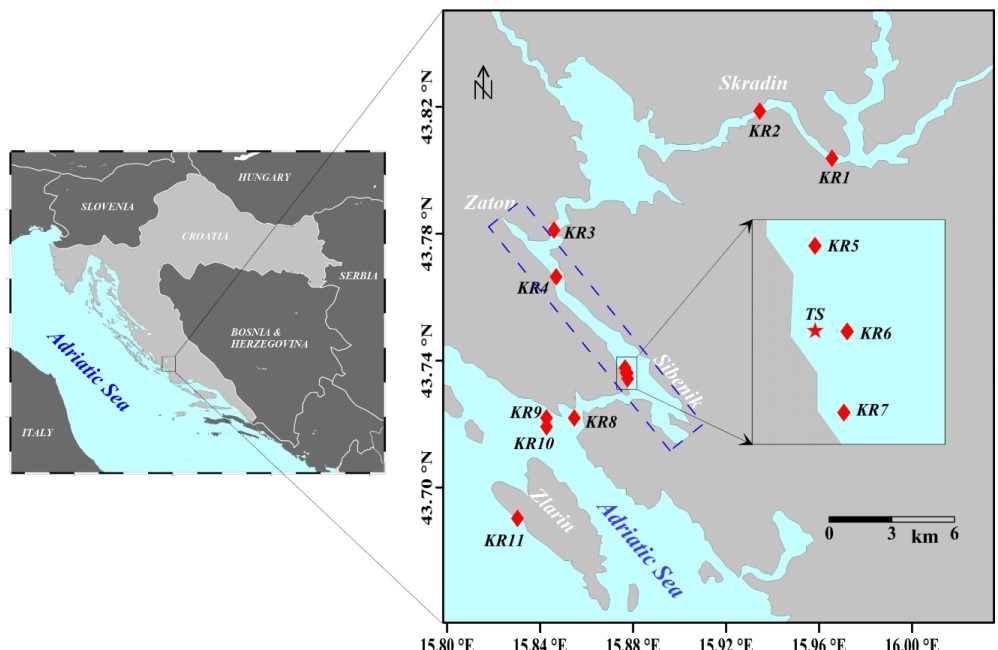

**Figure 1.** Map of the sampling area of the Krka River. Diamonds represents regular sampling stations and star represents the time series observation station. Dashed box represents the study area of interest for estimating SGD.





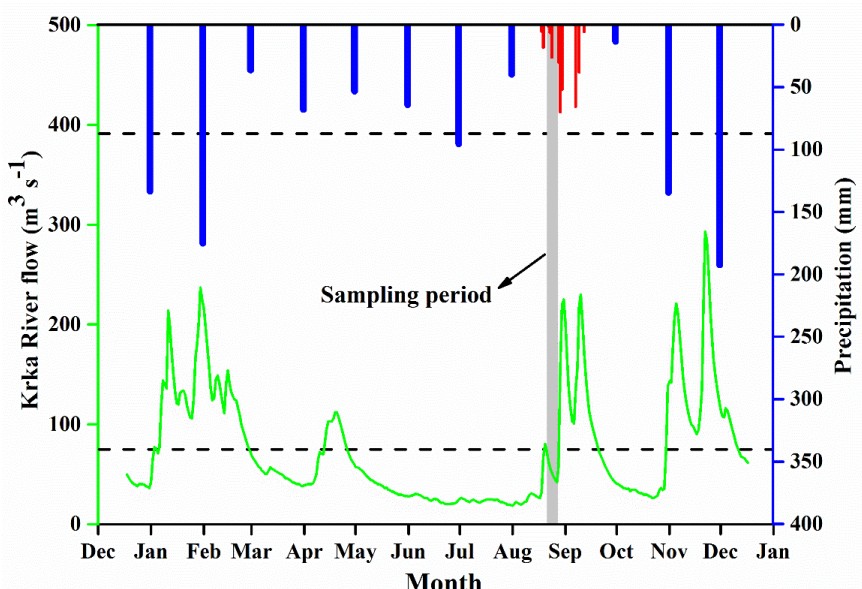

**Figure 2.** The Krka River flow and monthly precipitation in the city of Šibenik in 2014. Red bars represent the daily precipitation of the September 2014. Data from Šibenik meteo organization (http://www.sibenik-meteo.com).





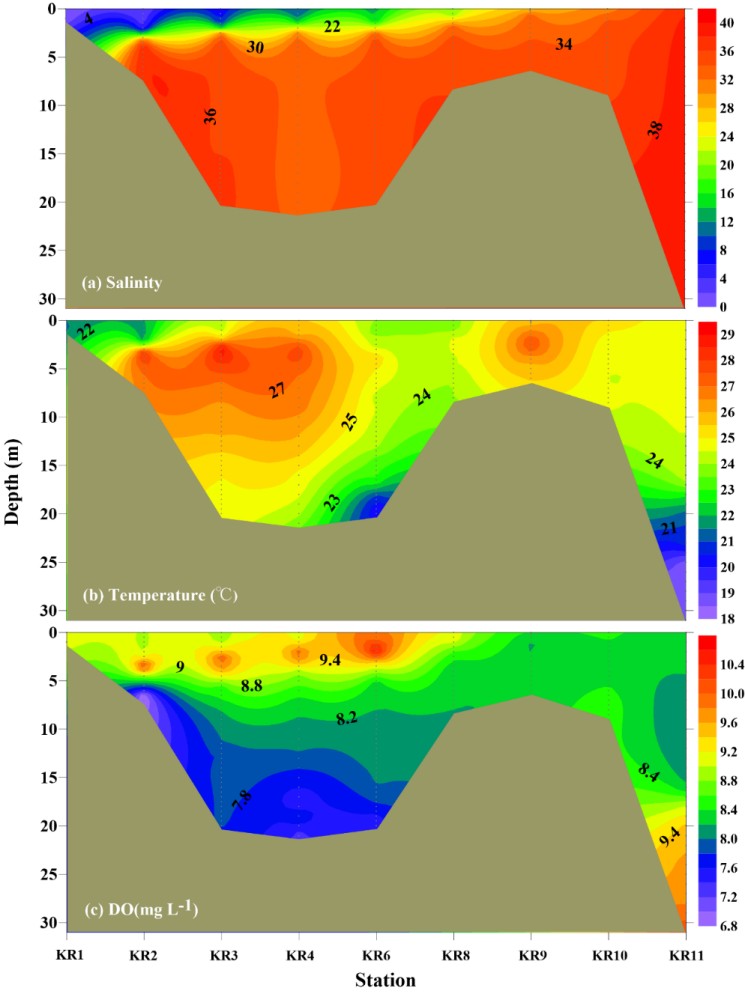

**Figure 3.** Vertical distributions of (a) salinity, (b) temperature and (c) dissolved oxygen (DO) from Krka River, along the estuary up to the Adriatic Sea.



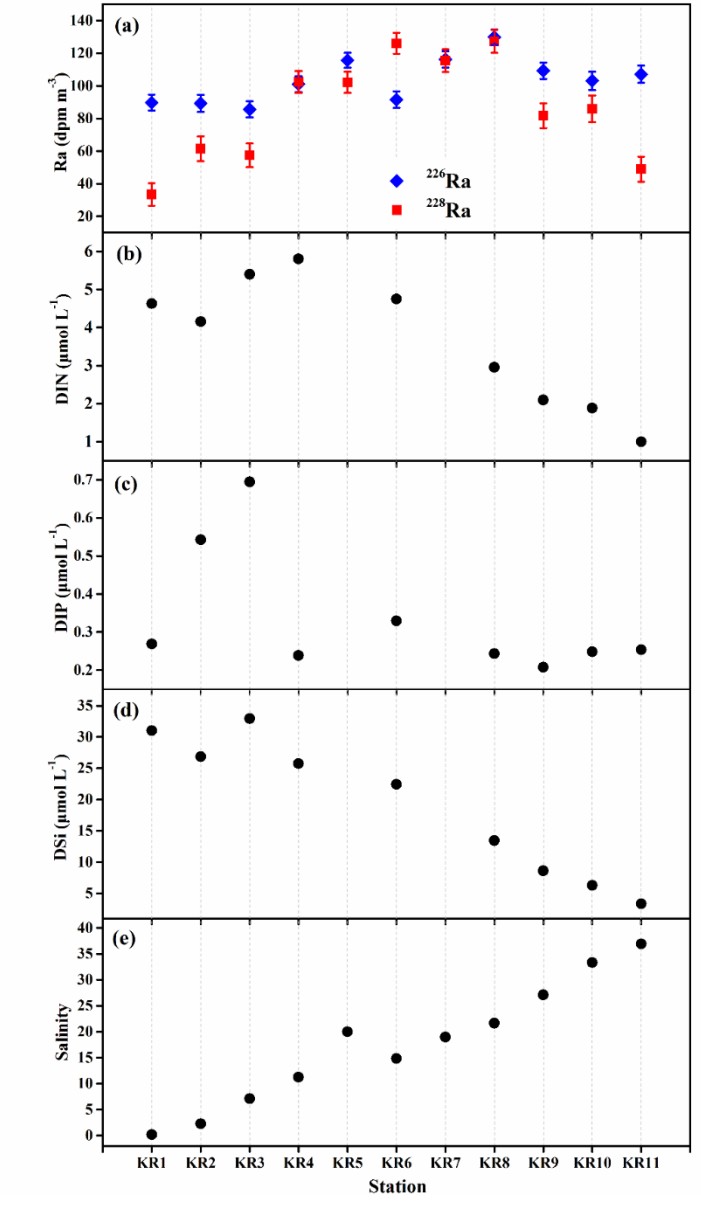

**Figure 4.** Distributions of (a) Ra, (b-d) nutrients and (e) salinity in the surface water of KRE.





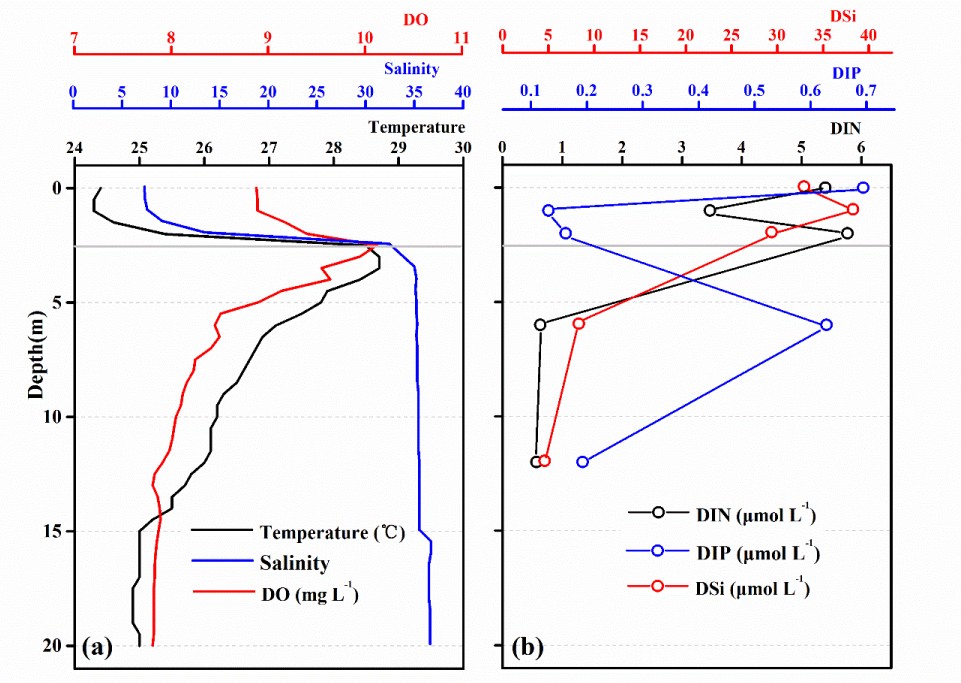

**Figure 5.** Vertical profiles of (a) hydrological parameters and (b) nutrient concentrations for station KR3.





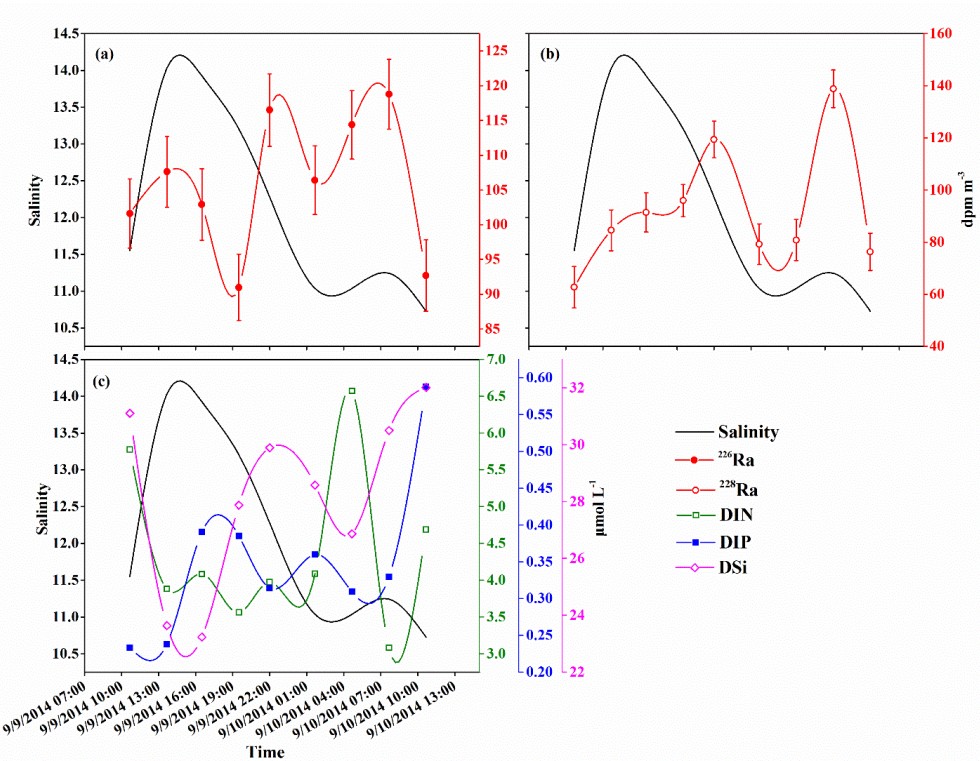

**Figure 6.** Salinity, Ra activities and nutrient concentrations variation in the surface water of the KRE during the time series observation.




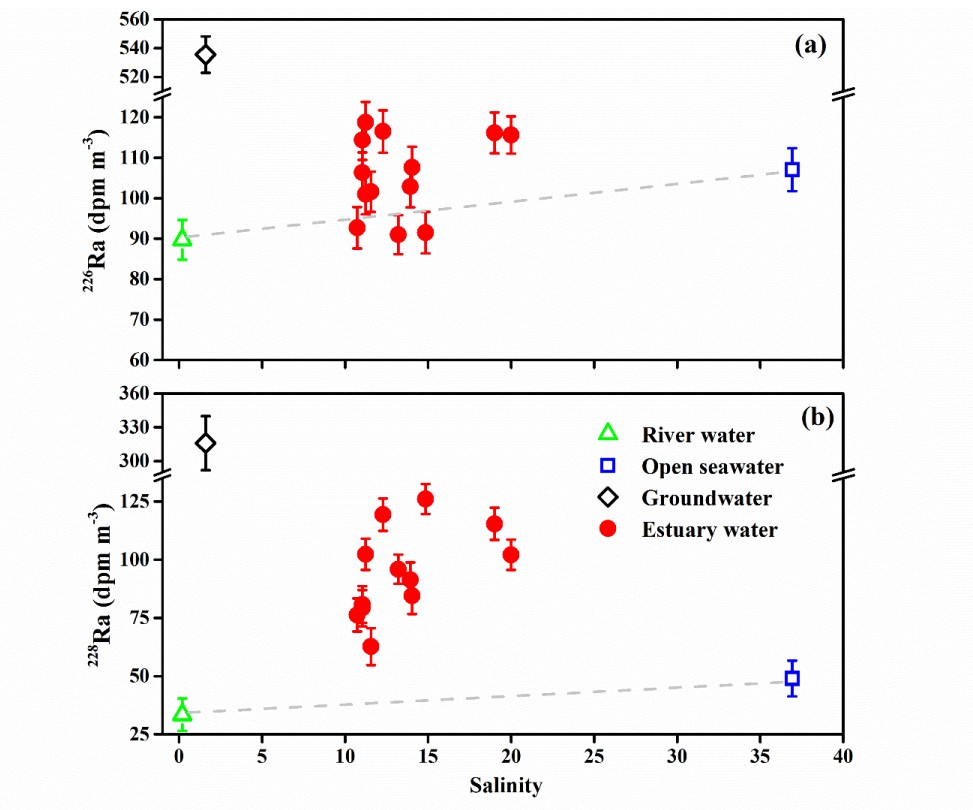

**Figure 7.** Plots of $^{226}$Ra and $^{228}$Ra activities versus salinity in the surface waters of the KRE.



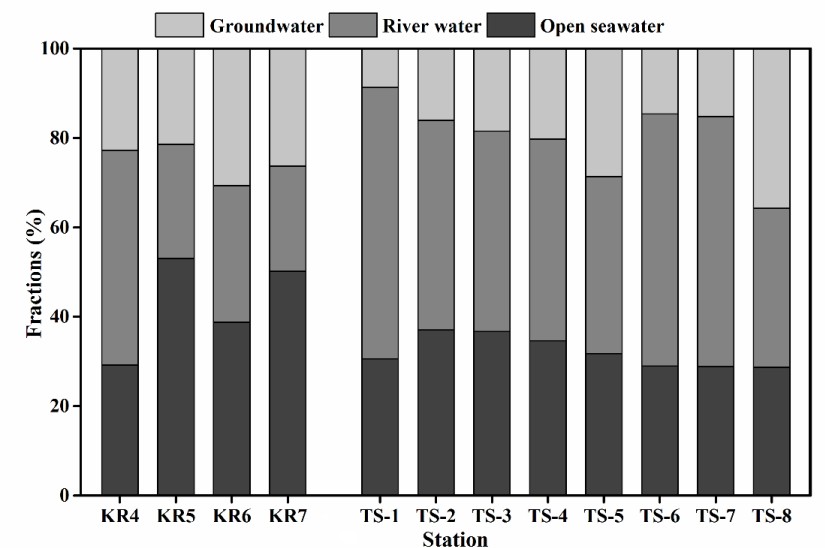

**Figure 8.** Fractions of groundwater, river water and open seawater in the surface water of KRE.





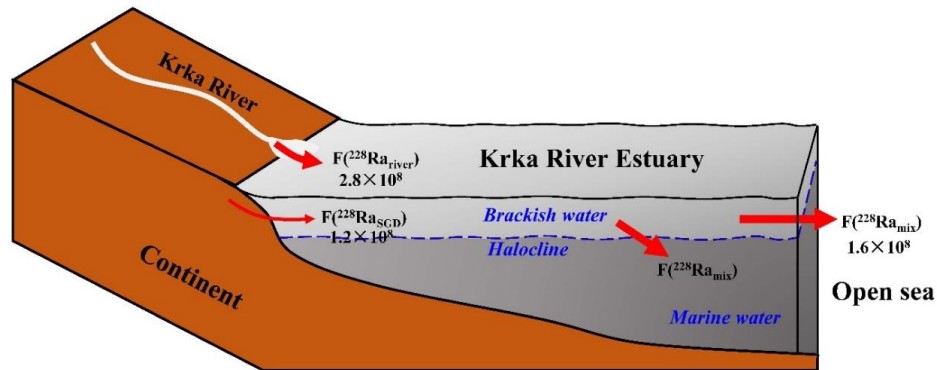

**Figure 9.** A schematic depiction of $^{228}$Ra mass balance (units in dpm d$^{-1}$) in the KRE surface layer.




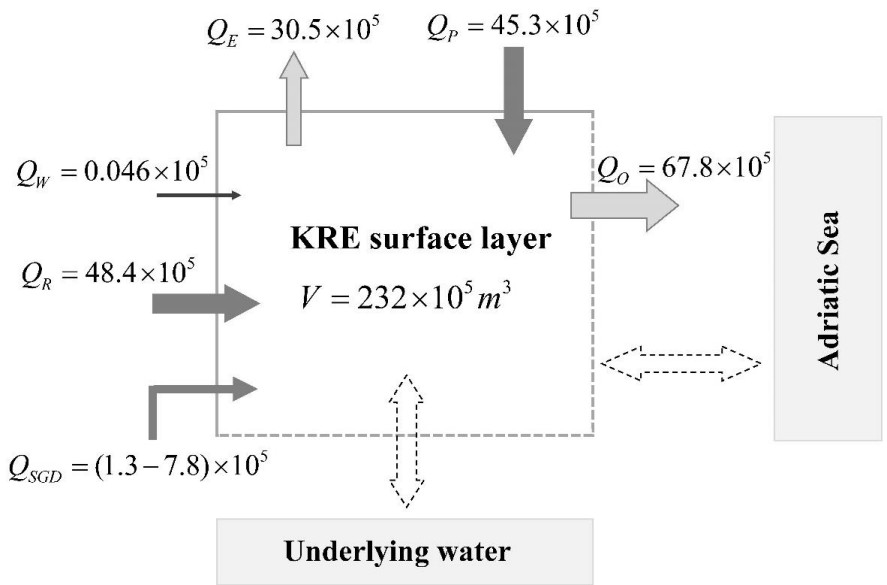

**Figure 10.** Water balance ($m^3$ $d^{-1}$) in the KRE surface layer.





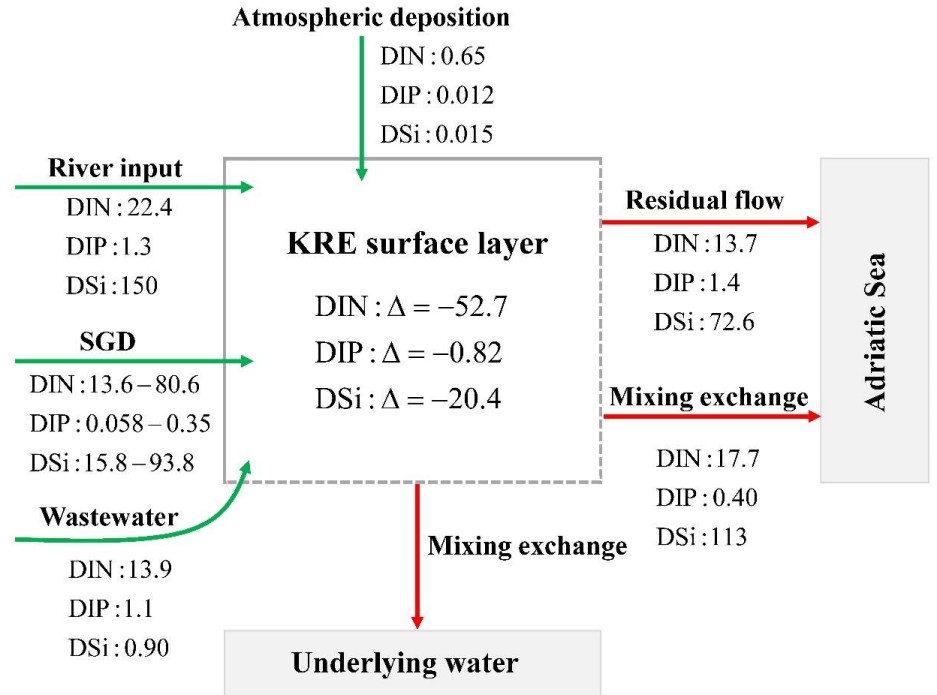

**Figure 11.** Nutrient budgets ($\times 10^3$ mol d$^{-1}$) in the KRE surface layer.