# Peer review of "226Ra and 228Ra in the stratified estuary of the Krka River (Adriatic Sea, Croatia): implications for submarine groundwater discharge and its derived nutrients"

_Biogeosciences, 2017_

## Referee Comment (RC1) · Anonymous Referee #1 · 30 Aug 2017

This study uses three different approaches to quantify SGD fluxes in the Krka River and estuary using Ra isotopes. The study is relevant as the understanding of groundwater nutrient inputs is of growing concern particularly in developed and modified systems like the Krka. The use of 3 different methodologies provides a unique approach to quantifying fluxes and I believe as such makes it relevant to a journal such as Biogeoscience Discussions. However I felt the scientific and presentation quality of the submitted manuscript are not to the standard needed for publication in this journal in its current state.

[Figure]
My main concern in the manuscript is the uncertainty around the Ra concentrations. As shown by the authors, there was a great range in Ra concentrations over the tidal cycle during their time series. Using a one off survey to gather Ra data for a mass balance introduces a large amount of uncertainty into how representative the survey was. More information is needed in the methods on how the survey was conducted (ie over 1 tidal cycle? over multiple days?) and how this might affect the Ra concentrations over the survey.

The authors state that trends in Ra activities were low in freshwaters, highest at the mouth and low in the estuary based on Figure 4. While there is a clear relationship between salinity and the sampling sites, I do not believe this is evident in Figure 4 particularly for 226Ra with highest 228Ra concentrations corresponding to near the lowest 226Ra. I believe, the error bars refer to analytical uncertainties from instrumentation rather than replicate measurements so do not give an indication a sampling variability at each site which would have been useful. A salinity vs Ra concentration plot would also have been useful. This plot is presented in Figure 7 and is said to include sampling "between Krka River water and open seawater" however the estuary samples presented fall in a very narrow salinity range (10-20) which do not correspond to those seen during the survey. Also there are more estuary measurements than sampling sites along the estuary. As such, it is not clear where this data comes from. This same comment applies to the author's interpretation of time series data in figure 6 (Page 4, line 34) as I do not believe a trend is evident. Statistical analysis would help quantify any trends. Added to this is the time series took place in a location where the authors note freshwater springs are present which is suggested to be the cause of higher Ra concentrations during the survey. This would dramatically skew flux calculations based on high point source Ra concentrations using the time series. Specific comments are below.

Specific comments: Overall, I found the manuscript contains numerous grammar and structural mistakes which at times made information difficult to follow. However, I be-

[Figure]

lieve this can be easily rectified by a thorough professional proof read and will not include those suggests below.

Page 4 Line 3. The authors variability in Ra concentrations due to hysteresis but do not demonstrate this. Statistical analysis or including the hysteresis analysis in a figure is needed to show that relationship as it is not clear.

Page 4 Line 5. As with the Ra concentrations, the nutrient trends are not clear from the figures. Correlation analysis would help clarify if such a relationship exists.

Page 6 line 1. As queried above, it is unclear where the estuary samples come from as they fall in a very narrow salinity range and do not come from the entire survey. Therefor the mass balance is based on a very narrow range of Ra concentrations in potentially a portion of the estuary receiving point sources of Ra (freshwater springs).

Page 7 Line 5. The episodic breakdown of boundary layers (ie Simpson et al Estuar. Coast. 1990 and Scully et al Estuar. Coast. 2005) needs to be discussed. This break down of the boundary layer may deliver high concentrations of Ra and nutrients to surface waters both spatially and temporally.

Page 8 Line 20. I believe this interpretation is limited as it does not include evapotranspiration, aquifer recharge or surface storage. Further to this, I believe the fact that this analysis contains significant uncertainties and it does not add to the main scientific story of the manuscript which is the use Ra isotopes to quantify SGD and SGD nutrient fluxes, I would omit.

Page 9 Line 23. The uncertainty in combine groundwater mass balance nutrient fluxes and average wastewater treatment plant fluxes needs to be discussed. Without knowing the time specific discharge of the plant and how it affected river and estuary nutrient concentration there are large assumptions in this model.

Figure 3. Using distance on the x axis would make the plot more easily interpreted. Including the sampling points in the plots would also help show the reader how accurate

the interpolation of data points was.

Figure 4. Using distance or salinity on the x axis would be more useful.

Figure 6. This was difficult to interpret due to how the legend was presented. Using titles on the y axis and a legend on each plot would help with this.

Figure 7. As above, unclear where the estuary samples are from as they have a narrow range and there are more of them than the survey. It could be problematic for the mass balance if the samples are from the time series due to point source Ra discharge.

Figure 8. It is unclear why only the middle section survey sites are included here.

―――――――――――――――――――――

---

## Referee Comment (RC2) · Anonymous Referee #2 · 5 Sep 2017

comments on "226Ra and 228Ra in the stratified estuary of the Krka River (Adriatic Sea, Croatia): implications for submarine groundwater discharge and its derived nutrients" by Liu et al.

The authors applied 3 models, the three-end member mixing model, the mass-balance model, and the time-series model to estimate the SGD flux to the Krka River estuary. In calculating the flushing time, the river discharge is not included, which should be considering the great river discharge of the Krka River during the investigation. In the three end-member mixing model and the mass-balance model, desorption of radium as a Ra

source in estuaries is not included or evaluated. In the time-series model, the assumptions are not substantiated. Ra and water depth at the time-series station change with tide/time. These changes are not only due to SGD, but also due to seawater into the estuary. The estuarine Ra background thus changes with time. All these changes are not considered in the time-series model. Only when these models are corrected, can the results and discussion be assessed and evaluated. There is not much discussion on implications for SGD-associated nutrients. It's better for the authors to turn to an English editor to go through all the texts.

Minor comments Page 1 Line 17: 'in tidal period' specify the time frame: 24 hours or 12 hours. Line 21: '9.5-38.3% to the total DSi flux' can't be taken as 'a major source'. Line 22: 'likely' is not proper to be used here. Quantitative results should allow the authors to determine whether SGD is a major source or not. This is no longer a possibility. Line 26: what is a river-dominated estuary? The sentence "It is the primary pathway..." is awkward. Rewrite it. Page 2 Line 23: what is tidal amplitude? Tidal range? Tidal height? Line 27: "have" to "has"; "transporting" to "transported". Line 34: "transect" to "transects". Page 3 Line 5: "24 hours-time" to "24-hour time" Line 9: dissolved oxygen (DO) measured using a multi-parameter probe needs to be calibrated with DO measured using classic Winkler titration. Otherwise, DO data are questionable. Line 13: the pore size seems too big to do the filtration. Double check the pore size of the cartridge. Line 17: the sentence "the 228Ra activities..." needs to be revised. Line 24: Briefly explain the method used to measure these nutrients. Provide the detection limits of these nutrients. Line 28: "investigated" to "investigation". Fig. 5 Line 15: "It was particularly pronounced for 228Ra that had lower effect than in the open Adria Sea". What effect? Line 16-18: how about adsorption of radium from particles as a source? It is not included in your three end-member mixing model and its effect should be evaluated. Fig. 6 Line 3: give the values of each fraction to support the statement "the fraction of the river water was higher than those of the open seawater and groundwater". Line 4: "lower changes" is an awkward phrase. Is 28-37% the change in the fraction or the fraction? Line 10: This physical model is not proposed by

Moore et al. (2006). Cite the original paper. Moreover, the model used here is suitable for estuaries with low river discharge, so river discharge is not considered. In your case the river discharge is not negligible and should be included in calculating the flushing time. Refer to the original paper for the proper formula to use here. Line 27: Every parameter on the right hand side of Eq. (8) is given as an average value with a standard deviation. How is the value on the left hand side of Eq. (8) calculated to be a range of values? Page 7 Line 6: The sentence is broken. Line 12: Again in Eq. (10) desorption of radium from particles is a source of dissolved radium that needs to be included in the mass-balance model. Line 27: the estuarine background changes with tides. When a minimum is chosen to be the estuarine background, an overestimate of SGD may result. Page 8 Line 1-3: the time-series Ra activity changes with tide. I suspect that the surface layer depth changes also with tide (the observation of water depth at the time-series station will verify this). Then the excess Ra inventory calculated with a constant water depth is not appropriate. Figure 1. Groundwater sampling stations are not shown. Figure 7. Only one groundwater point is shown. From Table 1 two groundwater samples were collected for Ra and both should be shown here.

---

## Author Comment (AC1) · 8 Nov 2017

**Response to Referee #1:**

Dear Referee, we appreciate your insightful and thorough comments and suggestions according to which we improved our revised MS carefully. Thank you for your time and critical evaluation and enclosed please kindly find our responses (written in blue) as follows.

**General comments:**

(1) My main concern in the manuscript is the uncertainty around the Ra concentrations. As shown by the authors, there was a great range in Ra concentrations over the tidal cycle during their time series. Using a one off survey to gather Ra data for a mass balance introduces a large amount of uncertainty into how representative the survey was. More information is needed in the methods on how the survey was conducted (ie over 1 tidal cycle? over multiple days?) and how this might affect the Ra concentrations over the survey.

**RESPONSE:**

Thank you for raising these important points. In this manuscript, the uncertainty of the Ra concentrations came from the errors of sampling, treatment and measurement. Actually, the dominant error was from Ra measurement. Using HPGe Gamma spectrometry (well-type, ORTEC, GWL-120-15-XLB-AWT), the error of $^{226}$Ra and $^{228}$Ra can be controlled below 20 % (Du et al., 2013; Wang et al., 2014, 2016), which was the similar case in other labs as well (e.g. Kim et al., 2005; Liu et al., 2012).

The time series observation was conducted by sampling at every 3 hours in a continuous 24-hour period over one complete tidal cycle. It is a very common method to evaluate the submarine groundwater discharge (SGD) by the time series observation (e.g. Peterson et al., 2008; Garcia-Orellana et al., 2010; Wang et al., 2016). During the time series observation, the tide variation can result in the change of Ra activity in the water column,because the water component is changing. At the low tide, weaker intrusion of the open seawater with low Ra concentration occurred along with the observation of higher percentage of submarine groundwater with higher Ra concentration. At the high tide, the situation was opposite, and hence the high activity of Ra was observed at low tide, whereas low Ra activity was observed at high tide.

Moreover, some information about this time series observation has been added in the Sampling strategy and Results and discussion sections in the revised manuscript as follows:

"In the KRE, we also conducted a continuous 24-hour time series observation over a complete tidal cycle by sampling the surface water at time-series (TS) station

every three hours (Figure 1), which has been widely used to evaluate the SGD (e.g., Peterson et al., 2008; Garcia-Orellana et al., 2010; Wang et al., 2016)."

"Activities of $^{226}$Ra and $^{228}$Ra showed similar opposite trend with respect to salinity due to the tidal variation (Figures 6a and 6b). At the low tide, weaker intrusion of the open seawater with low Ra concentration occurred along with the observation of higher percentage of submarine groundwater with higher Ra concentration. At the high tide, the situation was opposite, and hence the high activity of Ra was observed at low tide, whereas low Ra activity was observed at high tide."

Du, J. Z., Moore, W. S., Hsh, H. F., Wang, G. Z., Scholten, J., Henderson, P., Men, W., Rengarajan, R., Sha, Z. J., and Jiao, J. J.: Inter-comparison of radium analysis in coastal sea water of the Asian region, Mar, Chem., 156, 138-145, doi: 10.1016/j.marchem.2013.04.008, 2013.

Garcia-Orellana, J., Cochran, J. K., Bokuniewicz, H., Yang, S., and Beck, A. J.: Time-series sampling of $^{223}$Ra and $^{224}$Ra at the inlet to Great South Bay (New York): a strategy for characterizing the dominant terms in the Ra budget of the bay, J. Environ. Radioact., 101(7), 582-588, doi:10.1016/j.jenvrad.2009.12.005, 2010.

Kim, G., Ryu, J. W., Yang, H. S. and Yun, S. T.: Submarine groundwater discharge (SGD) into the Yellow Sea revealed by $^{228}$Ra and $^{226}$Ra isotopes: implications for global silicate fluxes, Earth Planet. Sci. Lett., 237(1), 156-166, doi: 10.1016/j.epsl.2005.06.011, 2005

Liu, Q., Dai, M., Chen, W., Huh, C.A., Wang, G., Li, Q. and Charette, M.A.: How significant is submarine groundwater discharge and its associated dissolved inorganic carbon in a river-dominated shelf system?, Biogeosciences, 9(5), 1777-1795, doi: 10.5194/bg-9-1777-2012, 2012.

Peterson, R. N., Burnett, W. C., Taniguchi, M., Chen, J., Santos, I. R., and Ishitobi, T.: Radon and radium isotope assessment of submarine groundwater discharge in the Yellow River delta, China. J. Geophys. Res.: Oceans, 113(C9), doi:10.1029/2008JC004776, 2008.

Wang, X., Du, J., Ji, T., Wen, T., Liu, S. and Zhang, J.: An estimation of nutrient fluxes via submarine groundwater discharge into the Sanggou Bay-a typical multi-species culture ecosystem in China, Mar. Chem., 167, 113-122, doi:10.1016/j.marchem.2014.07.002, 2014.

Wang, X. and Du, J.: Submarine groundwater discharge into typical tropical lagoons: A case study in eastern Hainan Island, China, Geochem. Geophys. Geosyst., 17(11), 4366-4382, doi:10.1002/2016GC006502, 2016.

(2) The authors state that trends in Ra activities were low in freshwaters, highest at the mouth and low in the estuary based on Figure 4. While there is a clear relationship between salinity and the sampling sites, I do not believe this is evident in Figure 4 particularly for 226Ra with highest 228Ra concentrations corresponding to near the lowest 226Ra. I believe, the error bars refer to analytical uncertainties from instrumentation rather than replicate measurements so do not give an indication a sampling variability at each site which would have been useful. A salinity vs Ra concentration plot would also have been useful. This plot is presented in Figure 7 and is said to include sampling "between Krka River water and open seawater" however the estuary samples presented fall in a very narrow salinity range (10-20) which do not correspond to those seen during the survey. Also there are more estuary measurements than sampling sites along the estuary. As such, it is not clear where this data comes from. This same comment applies

to the author's interpretation of time series data in figure 6 (Page 4, line 34) as I do not believe a trend is evident. Statistical analysis would help quantify any trends. Added to this is the time series took place in a location where the authors note freshwater springs are present which is suggested to be the cause of higher Ra concentrations during the survey. This would dramatically skew flux calculations based on high point source Ra concentrations using the time series.

**RESPONSE:**

In general, the Ra isotopes are conservative tracers, and their conservative behavior is more evident in the estuaries. In the freshwater, Ra isotopes are adsorbed on the suspended particles, then following the salinity increase in the estuary, Ra concentration increases due to the release of Ra from suspended particles or other Ra source input in low and middle range of salinity (below ~20). In higher salinity (over ~20) zone, owing to mixing with the open seawater, the Ra activity in seawater decreased with the rise of salinity, as the open seawater in general has notably lower Ra activity (Rutgers van der Loeff et al., 2003). Such case is very common in the estuarine zone, which is termed as inversed V type (e.g., Moore and Krest, 2004; Liu et al., 2012). It is shown in Figure 4, in which the $^{228}$Ra variation trend is more observable than $^{226}$Ra, because the $^{228}$Ra (half-life 5.7 yrs) in the freshwater and open sea water is much lower than that in the estuary relative to $^{226}$Ra (half-life 1600 yrs) and these result in less effects from freshwater and open sea water for $^{228}$Ra, thus the inversed V type variation trend of $^{228}$Ra was more evident. Our results are in agreement with earlier studies (e.g., Beck et al., 2007; Rengarajan and Sarma, 2015).

In this manuscript, the Ra error bars really refers to analytical uncertainties from instrumentation (HPGe Gamma spectrometry) rather than replicate measurements, and the measurement error bars are widely used for Ra worldwide (e.g., Liu et al., 2012; Rodellas et al., 2015).

Figure 7 shows more measurements than sampling sites in the estuary. We apologize that the presentation was not clear in the original manuscript. Actually the time series observation data are also included into the Figure 7. Besides that, only the estuarine data are plotted in Figure 7 and because of that, the salinity falls into the narrow range of 11.2-19. Since, the survey was conducted from the freshwater end-member to the open sea end-member, we have revised Figure 7 which now includes all the data from the freshwater end-member to the open sea end-member.

Figure 6 shows that the Ra activity has a negative correlation with salinity (r=-0.55, p=0.079 for $^{226}$Ra and r=-0.60, p=0.057 for $^{228}$Ra). This is due to the fact that open seawater has low Ra activity whilst the submarine groundwater inputs have high Ra activities.

We used only the high point source Ra concentrations of the time series observation to calculate the SGD flux. In fact, all the Ra data of the time series observation are used as shown by equation (12). That is why we can calculate the continuous variation of SGD flux at each time point during the tidal cycle, and obtain an integrated SGD flux range using the time series observation.

Finally, we corrected the concerned parts to clarity our presentation in the revised manuscript.

Beck, A.J., Rapaglia, J.P., Cochran, J.K. and Bokuniewicz, H.J.: Radium mass-balance in Jamaica Bay, NY: evidence for a substantial flux of submarine groundwater, Mar. Chem., 106(3), 419-441, doi: 10.1016/j.marchem.2007.03.008, 2007.

Liu, Q., Dai, M., Chen, W., Huh, C.A., Wang, G., Li, Q. and Charette, M.A.: How significant is submarine groundwater discharge and its associated dissolved inorganic carbon in a river-dominated shelf system?, Biogeosciences, 9(5), 1777-1795, doi: 10.5194/bg-9-1777-2012, 2012.

Moore, W.S. and Krest, J.: Distribution of $^{223}$Ra and $^{224}$Ra in the plumes of the Mississippi and Atchafalaya Rivers and the Gulf of Mexico. Mar. Chem., 86(3), 105-119, doi: 10.1016/j.marchem.2003.10.001, 2004.

Rengarajan, R. and Sarma, V. V. S. S.: Submarine groundwater discharge and nutrient addition to the coastal zone of the Godavari estuary, Mar. Chem., 172, 57-69, doi:10.1016/j.marchem.2015.03.008, 2015.

Rodellas, V., Garcia-Orellana, J., Masqué, P., Feldman, M., and Weinstein, Y.: Submarine groundwater discharge as a major source of nutrients to the Mediterranean Sea, Proc. Natl. Acad. Sci., 112(13), 3926-3930, doi:10.1073/pnas.1419049112, 2015.

Rutgers van der Loeff, M., Kühne, S., Wahsner, M., Höltzen, H., Frank, M., Ekwurzel, B., Mensch, M., Rachold, V.: $^{228}$Ra and $^{226}$Ra in the Kara and Laptev Seas, Cont. Shelf Res., 23(1), 113-124, doi:10.1016/S0278-4343(02)00169-3, 2003.

**Specific comments:**

(1) Overall, I found the manuscript contains numerous grammar and structural mistakes which at times made information difficult to follow. However, I believe this can be easily rectified by a thorough professional proof read and will not include those suggests below.

**RESPONSE:** We followed the advice of the referee, and the whole manuscript is proof read by the professor who is a native English speaker.

(2) Page 4 Line 3. The authors variability in Ra concentrations due to hysteresis but do not demonstrate this. Statistical analysis or including the hysteresis analysis in a figure is needed to show that relationship as it is not clear.

**RESPONSE:** We agree with the referee on this point. The Ra sources, especially SGD could not respond to the tidal variation so fast, which results in hysteresis (e.g., Sadat-Noori et al., 2015). When the hysteresis was taken into consideration, the correlation analysis showed that Ra concentrations and salinity had a

considerable negative correlation (r=-0.55 for $^{226}$Ra and r=-0.60 for $^{228}$Ra) despite of the non-significant p-values (p=0.079 for $^{226}$Ra p=0.057 for $^{228}$Ra). We have added this point in the revised manuscript.

Sadat-Noori, M., Santos, I. R., Sanders, C. J., Sanders, L. M. and Maher, D. T.: Groundwater discharge into an estuary using spatially distributed radon time series and radium isotopes, J. Hydrol., 528,703-719, doi:10.1016/j.jhydrol.2015.06.056, 2015.

(3) Page 4 Line 5. As with the Ra concentrations, the nutrient trends are not clear from the figures. Correlation analysis would help clarify if such a relationship exists.

**RESPONSE:** We followed the advice of the referee. Similar to that of Ra activity, DIP and DSi variations had an opposite trend to salinity with a small hysteresis effect observed. There was no obvious variation correlation between DIN and salinity. Therefore, as with the Ra, with the hysteresis effect considered, we performed the correlation analysis, which showed that none of the nutrients had a significant correlation with salinity (r=-0.44, p=0.123 for DIP, r=-0.43, p=0.147 for DSi and r=0.16, p=0.341 for DIN).

(4) Page 6 line 1. As queried above, it is unclear where the estuary samples come from as they fall in a very narrow salinity range and do not come from the entire survey. Therefore, the mass balance is based on a very narrow range of Ra concentrations in potentially a portion of the estuary receiving point sources of Ra (freshwater springs).

**RESPONSE:** As shown by the blue dashed line box in Figure 1, the Ra data in three end-member mixing model includes only time series samples (TS-1~9) and the samples from the estuary (KR4~7). Only the middle zone of the estuary and its surface waters above the halocline are presented in this figure.

In this survey, the salinity of the KRE in surface water was below 21.7 (Table 1), similar to the salinity of the Ra samples used in the three end-member mixing model and mass balance model. Therefore, we believe it is reasonable to calculate the SGD flux using only the estuarine samples. In consideration of morphology and sediment properties, freshwater spring is assumed to be similar, because when the freshwater spring comes out it mixes with seawater immediately and usually the salinity is in the range of the KRE samples.

(5) Page 7 Line 5. The episodic breakdown of boundary layers (ie Simpson et al Estuar. Coast. 1990 and Scully et al Estuar. Coast. 2005) needs to be discussed. This break down of the boundary layer may deliver high concentrations of Ra and nutrients to surface waters both spatially and temporally.

**RESPONSE:** The Krka River Estuary is highly stratified and the breakdown of the boundary layer is less likely and less important than in some other estuaries. The boundary layer (i.e. halocline) is permanent in KRE, whilst the thickness and the steepness of the salinity gradient change seasonally and longitudinally from its head to mouth (Žutić and Legović, 1987; Legović, 1991; Cukrov et al., 2009). Figure 5 shows that the parameters are highly different above and below the halocline, and it is obvious that the halocline can significantly slow down the diffusion of nutrients from the underlying water to the surface layer water, in line with Legović et al. (1994). Thus, we omitted the impact of diffusion of Ra and nutrients through the halocline.

Cukrov, N., Mlakar, M., Cuculić, V., and Barišić, D.: Origin and transport of [238]U and [226]Ra in riverine, estuarine and marine sediments of the Krka River, Croatia, J. Environ. Radioact., 100(6), 497-504, doi:10.1016/j.jenvrad.2009.03.012, 2009.

Legović, T.:  Exchange of water in a stratified estuary with an application to Krka (Adriatic Sea), Mar. Chem., 32(2), 121-135, doi:10.1016/0304-4203(91)90032-R, 1991.

Legović, T., Žutić, V., Gržetić, Z., Cauwet, G., Precali, R., and Viličić, D.: Eutrophication in the Krka estuary, Mar. Chem., 46(1), 203-215, doi:10.1016/0304-4203(94)90056-6, 1994.

Žutić, V. and Legovic, T.: A film of organic matter at the fresh-water/sea-water interface of an estuary. Nat., 328(6131):612-614, doi:10.1038/328612a0, 1987.

(6) Page 8 Line 20. I believe this interpretation is limited as it does not include evapotranspiration, aquifer recharge or surface storage. Further to this, I believe the fact that this analysis contains significant uncertainties and it does not add to the main scientific story of the manuscript which is the use Ra isotopes to quantify SGD and SGD nutrient fluxes, I would omit.

**RESPONSE:** The water balance model includes evaporation as $Q_E$ term in Equation (13), but really neglects the net variation of water storage with time (i.e. over a tidal period) in this system, because this model is a classic method to build water balance in the system (Benduhn and Renard, 2004; Wang et al., 2015). It is a necessary step for establishing the following nutrients budgets and emphasizing the importance of the SGD-derived nutrients. For these reasons, we would like to keep these results in the revised manuscript.

Benduhn, F. and Renard, P.: A dynamic model of the Aral Sea water and salt balance. J. Mar. Syst., 47(1):35-50, doi:10.1016/j.jmarsys.2003.12.007, 2004.

Wang, X. Li, H., Jiao, J. J., Barry, D. A., Li, L., Luo, X., Wang, C., Wan, L., Wang, X., Jiang, X., Ma, Q., and Qu, W.: Submarine fresh groundwater discharge into Laizhou Bay comparable to the Yellow River flux, Sci. Rep., 5, 8814, doi:10.1038/srep08814, 2015.

(7) Page 9 Line 23. The uncertainty in combine groundwater mass balance nutrient fluxes and average wastewater treatment plant fluxes needs to be discussed. Without knowing the time specific discharge of the plant and how it affected river and estuary nutrient concentration there are large assumptions in this model.

**RESPONSE:** As the water balance model, the Land Ocean Interactions in the Coastal Zone (LOICZ) approach is extensively used to establish nutrients budgets in the estuaries (e.g. Gordon et al., 1996; Hung and Hung, 2003; Liu et al., 2009, 2011). This model is established under several assumptions, for example, the estuary is treated as a single box and is assumed to be at a steady state.

Hung, J.-J. and Hung, P.-Y.: Carbon and nutrient dynamics in a hypertrophic lagoon in southwestern Taiwan, J. Mar. Syst., 42, 97–114, doi:10.1016/S0924-7963(03)00069-1, 2003.

Gordon, D. C., Boudreau, P. R., Mann, K. H., Ong, J. E., Silvert, W. L., Smith, S. V., Wattayakorn, G., Wulff, F., and Yanagi, T.: LOICZ Biogeochemical Modelling Guidelines (vol 5), LOICZ Core Project, Netherlands Institute for Sea Research, 1996.

Liu, S. M., Hong, G. H., Zhang, J., Ye, X. W., and Jiang, X. L.: Nutrient budgets for large Chinese estuaries, Biogeosciences, 6(10), 2245-2263, doi:10.5194/bg-6-2245-2009, 2009.

Liu, S. M., Li, R. H., Zhang, G. L., Wang, D. R., Du, J. Z., Herbeck, L. S., Zhang, J., and Ren, J. L.: The impact of anthropogenic activities on nutrient dynamics in the tropical Wenchanghe and Wenjiaohe Estuary and Lagoon system in East Hainan, China, Mar. Chem., 125(1), 49-68, doi:10.1016/j.marchem.2011.02.003, 2011.

(8) Figure 3. Using distance on the x axis would make the plot more easily interpreted. Including the sampling points in the plots would also help show the reader how accurate the interpolation of data points was.

**RESPONSE:** We agree with referee and have corrected it in the revised manuscript.

(9) Figure 4. Using distance or salinity on the x axis would be more useful.

**RESPONSE:** We have followed the advice of referee and have revised it as distance on the x axis.

(10) Figure 6. This was difficult to interpret due to how the legend was presented. Using titles on the y axis and a legend on each plot would help with this.

**RESPONSE:** We have followed the advice of referee and corrected it.

(11) Figure 7. As above, unclear where the estuary samples are from as they have a narrow range and there are more of them than the survey. It could be problematic for the mass balance if the samples are from the time series due to point source Ra discharge.

**RESPONSE:** The samples in Figure 7 include time series samples (TS-1~9) and the samples from the estuary (KR4~7).

(12) Figure 8. It is unclear why only the middle section survey sites are included here.

**RESPONSE:** The study was focused on the Krka River Estuary that was shown in the blue dashed box in Figure 1. We displayed the data for this main part of the estuary, namely the middle section of KRE presented at Figure 8.

---

## Author Comment (AC2) · 8 Nov 2017

**Response to Referee #2:**

Dear Referee, we appreciate your insightful and thorough comments and suggestions, according to which we improved our revised MS carefully. Thank you for your time and critical evaluation and enclosed please kindly find our responses (written in blue) as follows.

**General comments:**

The authors applied 3 models, the three-end member mixing model, the mass-balance model, and the time-series model to estimate the SGD flux to the Krka River estuary. In calculating the flushing time, the river discharge is not included, which should be considering the great river discharge of the Krka River during the investigation. In the three end-member mixing model and the mass-balance model, desorption of radium as Ra a source in estuaries is not included or evaluated. In the time-series model, the assumptions are not substantiated. Ra and water depth at the time-series station change with tide/time. These changes are not only due to SGD, but also due to seawater into the estuary. The estuarine Ra background thus changes with time. All these changes are not considered in the time-series model. Only when these models are corrected, can the results and discussion be assessed and evaluated. There is not much discussion on implications for SGD-associated nutrients. It's better for the authors to turn to an English editor to go through all the texts.

**RESPONSE:**

Thank you very much for your comments. We did a major revision of our manuscript according to your suggestions and hope that we achieved a desired scientific quality both in using the most appropriate methods and in presentation of our results. We will explain all the details considering the improvements according to your comments.

Firstly, we have added the river discharge term into flushing time model and used the equation from Sanford et al. (1992) to re-calculate the flushing time (Equation 7).

Secondly, considering the low suspended particle matter concentration in the Krka River Estuary, in the three end-member mixing model and the mass-balance model, we have evaluated desorption of radium from particles and it appeared to be a negligible term. Therefore, we did not include it into the model. We agree that our presentation was unclear in the submitted original manuscript, and therefore we have corrected these in the revised manuscript.

Thirdly, in the time series model, we have chosen the minimum measured Ra activity of the time series observation as the background of the estuarine water. The estimated SGD was obtained by subtraction of background Ra activities from the measured Ra activities. So, we could exclude the other Ra sources variations.

The similar cases can be found in our previous work and other publications (Peterson et al., 2008; Wang et al., 2016).

Fourthly, we have extended the discussion on implications for SGD-associated nutrients in the revised manuscript.

Finally, the whole manuscript is edited by the professor who is a native English speaker.

Sanford, L. P., Boicourt, W. C., and Rives, S. R.: Model for estimating tidal flushing of small embayments. J. Waterw., Port, Coastal, Ocean Eng., 118(6):635-54, doi: 10.1061/(ASCE)0733-950X(1992)118:6(635), 1992.

Peterson, R. N., Burnett, W. C., Taniguchi, M., Chen, J., Santos, I. R., and Ishitobi, T.: Radon and radium isotope assessment of submarine groundwater discharge in the Yellow River delta, China. J. Geophys. Res.: Oceans, 113(C9), doi:10.1029/2008JC004776, 2008.

Wang, X. and Du, J.: Submarine groundwater discharge into typical tropical lagoons: A case study in eastern Hainan Island, China, Geochem. Geophys. Geosyst., 17(11), 4366-4382, doi:10.1002/2016GC006502, 2016.

**Minor comments:**

Page 1 Line 17: 'in tidal period' specify the time frame: 24 hours or 12 hours.

**RESPONSE:** It's 24 hours, and we have added it into the revised manuscript.

Line 21: '9.5-38.3% to the total DSi flux' can't be taken as 'a major source'.
**RESPONSE:** We have corrected it according to new calculations.

Line 22: 'likely' is not proper to be used here. Quantitative results should allow the authors to determine whether SGD is a major source or not. This is no longer a possibility.

**RESPONSE:** We have corrected it according to new calculations.

Line 26: what is a river-dominated estuary? The sentence "It is the primary pathway: : :" is awkward. Rewrite it.

**RESPONSE:** We have rewritten that as "An estuary is the critical zone connecting the mainland and adjacent sea, and the primary region where continuous exchange of water and chemical components between land and sea/ocean occurs."

Page 2 Line 23: what is tidal amplitude? Tidal range? Tidal height?

**RESPONSE:** It is the tidal range and we have corrected it in the revised manuscript.

Line 27: "have" to "has"; "transporting" to "transported". Line 34: "transect" to "transects". Page 3 Line 5: "24 hours-time" to "24-hour time"

**RESPONSE:** We have corrected them.

Line 9: dissolved oxygen (DO) measured using a multi-parameter probe needs to be calibrated with DO measured using classic Winkler titration. Otherwise, DO data are questionable.

**RESPONSE:** The multi parametric probe has been calibrated using the Winkler titration. We have found a significant linear regression ($R^2$=0.791, n=88, df=86, p<0.0001) between DO values from the Winkler method and the multi parametric probe. Winkler method values were higher and considered as accurate, therefore all measurements from the probe were recalculated to the Winkler values via the equation of the regression line: Winkler DO=0.972*Probe DO+0.507 mg $O_2$/l. The conversion factor 1 mL $O_2$/L=1.42903 mg $O_2$/L (Owens and Millard 1985, Garcia and Gordon 1992) was employed. As we did not expect hypoxic samples in our study, i.e. the most accurate measurements by the Winkler method were not a priority, we used calibrated multi-parameter probe.

Garcia, H.E. and Gordon, L.I.: Oxygen solubility in seawater: Better fitting equations, Limnol. Oceanogr., 37(6),1307-1312, doi: 10.4319/lo.1992.37.6.1307, 1992.

Owens, W.B. and Millard Jr, R.C.: A new algorithm for CTD oxygen calibration, J. Phys. Oceanogr., 15(5), 621-631, doi: 10.1175/1520-0485(1985)015<0621:ANAFCO>2.0.CO;2, 1985.

Line 13: the pore size seems too big to do the filtration. Double check the pore size of the cartridge.

**RESPONSE:** We made a mistake. The pore size is 0.5 µm and we have corrected it in the revised manuscript.

Line 17: the sentence "the 228Ra activities: : :" needs to be revised.

**RESPONSE:** We have revised it as follows "while [228]Ac (338 keV and 911 keV peaks) was used for measuring [228]Ra activity."

Line 24: Briefly explain the method used to measure these nutrients. Provide the detection limits of these nutrients.

**RESPONSE:** The concentrations of nitrate ($NO_3^-$), nitrite ($NO_2^-$), $NH_4^+$, reactive orthosilicates ($SiO_4^{4-}$, hereafter termed DSi), orthophosphate ($PO_4^{3-}$, hereafter

termed dissolved inorganic phosphorus, i.e. DIP), were determined as described in Strickland and Parsons (1972) and Grasshoff et al. (1983), using a spectrophotometer (PerkinElmer Lambda15) combining 1 cm and 10 cm cuvettes, as needed. The detection limits and reproducibility for nutrients were as follows: 0.05 and 0.025 µmol L$^{-1}$ for $NO_3^-$; 0.01 and 0.01 µmol L$^{-1}$ for $NO_2^-$; 0.1 and 0.098 µmol L$^{-1}$ for $NH_4^+$; 0.1 and 0.06 µmol L$^{-1}$ for $SiO_4^{4-}$ and DIP 0.03 and 0.03 µmol L$^{-1}$ for DIP. We have added these into the revised manuscript.

Grasshoff, K., Kremling, K., and Ehrhardt, M.: Methods of seawater analysis, 2nd ed. Weinheim: Verlag Chemie GmbH, 1983.

Strickland, J. D. and Parsons, T. R.: A practical handbook of seawater analysis, 2nd ed. Ottawa, Bulletin of the Fisheries Research Board of Canada, 1972.

Line 28: "investigated" to "investigation".

**RESPONSE:** We have Corrected it.

Fig. 5 Line 15: "It was particularly pronounced for 228Ra that had lower effect than in the open Adria Sea". What effect?

**RESPONSE:** Due to the shorter half-life of $^{228}$Ra, in the open Adriatic Sea the $^{228}$Ra activity is much lower than that in the estuary relative to the $^{226}$Ra activity. Therefore, using $^{228}$Ra to establish the three end-member mixing model is more appropriate due to its lower mixing effect from the open sea.

Line 16-18: how about adsorption of radium from particles as a source? It is not included in your three end-member mixing model and its effect should be evaluated.

**RESPONSE:** As we stated above, we have evaluated desorption of radium from particles, which appeared to be a negligible term, so we did not include it into the model. Our presentation was unclear in the original manuscript, and we have corrected that in the revised manuscript.

Fig. 6 Line 3: give the values of each fraction to support the statement "the fraction of the river water was higher than those of the open seawater and groundwater".

**RESPONSE:** We have corrected it.

Line 4: "lower changes" is an awkward phrase. Is 28-37% the change in the fraction or the fraction?

**RESPONSE:** It means the smaller variation range. We have corrected it.

Line 10: This physical model is not proposed by Moore et al. (2006). Cite the original paper. Moreover, the model used here is suitable for estuaries with low river discharge, so river discharge is not considered. In your case the river discharge is not negligible and should be included in calculating the flushing time. Refer to the original paper for the proper formula to use here.

**RESPONSE:** We have added the river discharge term into the model and used the equation from Sanford et al. (1992) to re-calculate the flushing time as mentioned above.

Line 27: Every parameter on the right hand side of Eq. (8) is given as an average value with a standard deviation. How is the value on the left hand side of Eq. (8) calculated to be a range of values?

**RESPONSE:** We obtained the value with a standard deviation first, and then presented it as a range. Since we used three methods to estimate the SGD flux, we believe that using a range that covers all the values is accurate, and the values presented in this style were reported only for the SGD flux and its derived nutrients in our manuscript.

Page 7 Line 6: The sentence is broken.

**RESPONSE:** We have corrected it.

Line 12: Again in Eq. (10) desorption of radium from particles is a source of dissolved radium that needs to be included in the mass-balance model.

**RESPONSE:** We have corrected that in the revision as mentioned above.

Line 27: the estuarine background changes with tides. When a minimum is chosen to be the estuarine background, an overestimate of SGD may result.

**RESPONSE:** The Ra background generally includes riverine inputs, desorption from particles and mixing with an open sea, in which the mixing term changes with tides' height variation; and the minimum in the time-series observation also includes the SGD source. The variation range of mixing with open sea is much lower than the SGD source (Figure 8). Therefore, for each measured Ra activity, the minimum is subtracted out and that results in a conservative estimation.

Page 8 Line 1-3: the time-series Ra activity changes with tide. I suspect that the surface layer depth changes also with tide (the observation of water depth at the time-series station will verify this). Then the excess Ra inventory calculated with a constant water depth is not appropriate.

**RESPONSE:** The water depth varied within the time series observation, but the amplitude was small (no more than 0.3 m here), so we neglected the variation and assumed it to be a constant water depth, as also used by Peterson et al. (2008) and Wang et al. (2016).

Peterson, R. N. Burnett, W. C., Taniguchi, M., Chen, J., Santos, I. R., and Ishitobi, T.: Radon and radium isotope assessment of submarine groundwater discharge in the Yellow River delta, China. J. Geophys. Res.: Oceans, 113(C9), doi:10.1029/2008JC004776, 2008.

Wang, X. and Du, J.: Submarine groundwater discharge into typical tropical lagoons: A case study in eastern Hainan Island, China, Geochem. Geophys. Geosyst., 17(11), 4366-4382, doi:10.1002/2016GC006502, 2016.

Figure 1. Groundwater sampling stations are not shown.

**RESPONSE:** We have corrected it.

Figure 7. Only one groundwater point is shown. From Table 1 two groundwater samples were collected for Ra and both should be shown here.

**RESPONSE:** We have corrected it.